# Continuous Symmetry Discovery and Enforcement for Image Data

## Abstract

Symmetry is an often-desired quality of machine learning models, leading, among other things, to more predictable model generalization. Continuous symmetry detection and enforcement for machine learning are two related topics that have recently been explored using the Lie derivative along vectors fields, which vector field approach has led to improved outcomes. However, though image data is replete with continuous symmetries under which image classifiers are meant to be invariant, the application of the Lie derivative for the detection and enforcement of continuous symmetries for image data remains underexplored. In this work, we derive vector field infinitesimal generators for various continuous symmetries for image data. We then use these generators to enforce continuous symmetries in image classifiers. We also demonstrate vector field symmetry detection in image data, obtaining close similarity with the ground truth symmetry.

## 1 Introduction

Symmetry-informed machine learning is a field which has continued to accumulate interest, with previous work demonstrating its effectiveness in improving model performance (Lyle et al., 2020; Bergman, 2019; Craven et al., 2022; Tahmasebi & Jegelka, 2023; Ko et al., 2024). Within the context of image classification tasks, one is often concerned with learning neural networks which are invariant to pre-specified transformations of images. Although a common approach to building invariant models in practice is to explicitly augment the training data with transformed copies of images, recent work has been conducted with the goal of enforcing either equivariance or invariance in models without the need for data augmentation (Finzi et al., 2020). The general need to enforce symmetry in models without augmentation is due to two primary factors: (1) data augmentation can be computationally expensive (Mumuni & Mumuni, 2022); (2) data augmentation can only be applied if one knows or can discover the symmetry group explicitly. It is the second of these factors that is primarily applicable to image data, as common augmentations, such as rotations or adjusting the brightness, are relatively computationally inexpensive: nevertheless, employing augmentation for image data requires an explicit transformation of the data, while the transformation group may not be known *a priori*.

Recent work has shown that the Lie derivative along vector fields can be used to enforce and discover continuous symmetries in models quite generally, which approach can be computationally efficient (Otto et al., 2024; Shaw et al., 2024; 2025). However, it appears that, despite the many continuous symmetries for image data, previous work with regard to image data deals only with certain types of continuous transformations, such as planar affine transformations. In this work, we propose an approach to symmetry discovery and enforcement for image data using the Lie derivative along vector fields, thereby expanding the types of symmetries that can be considered.

Continuous symmetry enforcement for models has recently been accomplished using vector fields as a regularization term in the model loss function, similar to the framework of Physics-Informed Neural Networks (Shaw et al., 2025; Otto et al., 2024; Raissi et al., 2019). However, this approach, as currently defined, cannot directly be applied to symmetries for image data. This is because the approach requires that each datapoint reside in $\mathbb{R}^d$, where $d$ is the number of features: Although an image can be "flattened" so as to be readily interpretable as a vector in $\mathbb{R}^d$, common continuous image transformations such as those making use of convolutions, may not be recognizable in this feature space. Even when an image transformation is easily expressible in the flattened feature

space, it is often best practice to use convolutional layers when training on image data, highlighting the benefit of maintaining the integrity of the image structure. Therefore, the extension to image data is non-trivial and, as will be shown, is fruitful. As the vector field approach to model symmetry discovery and enforcement relies on the computation of the gradient of the model function, our extension to image data makes use of the gradient of the model with respect to an input image. The mathematical framework for this extension also relies on the notion of diagonal group actions.

**Our main contribution is summarized as follows**: we provide a mathematical framework for the extension of continuous symmetry discovery and enforcement for image data. We also provide experimental results which demonstrate the feasibility of implementing our method. Our results suggest that continuous symmetry enforcement in image classification using vector fields is comparable to augmentation and is applicable in cases in which augmentation cannot be used. Our results also suggest that model symmetry can be discovered and quantified using the Lie derivative.

## 2 RELATED WORK AND BACKGROUND

In this section, we discuss both related work and background information which is useful for understanding the context of our contributions. In Sections 2.1 and 2.2, we discuss previous work in symmetry discovery and enforcement, respectively. Section B contains an overview of vector fields and flows, being a somewhat new concept for machine learning, though we note that a similar discussion is contained in existing literature (Otto et al., 2024; Shaw et al., 2024). Section C gives a brief overview of infinitesimal generators of multi-parameter groups, and section 2.3 gives a brief overview of diagonal group actions and their infinitesimal generators.

### 2.1 RELATED WORK: SYMMETRY DISCOVERY

Early work on symmetry detection in machine learning focused primarily on detecting symmetry in image and video data (Rao & Ruderman, 1999; Sohl-Dickstein et al., 2010), where symmetries described by straight-line and rotational transformations were discovered. Other work has made strides in symmetry discovery by restricting the types of symmetries being sought. In one case, detection was limited to compact Abelian Lie groups (Cohen & Welling, 2014) and used for the purpose of learning disentangled representations. Another case uses meta-learning to discover any *finite* symmetry group (Zhou et al., 2021). Finite groups have also been used in symmetry discovery in representation learning (Anselmi et al., 2019). In physics-based applications of machine learning, a method was developed that can discover any classical Lie group symmetry (Forestano et al., 2023). Symmetry discovery of shapes has also been explored (Je et al., 2024).

Other work has focused on detecting affine transformation symmetries and encoding the discovered symmetries automatically into a model architecture. Some methods identify Lie algebra generators to describe the symmetries. For example, *augerino* (Benton et al., 2020) attempts to learn a distribution over augmentations, subsequently training a model with augmented data. The *Lie algebra convolutional network* (Dehmamy et al., 2021), which generalizes Convolutional Neural Networks in the presence of affine symmetries, uses infinitesimal generators represented as vector fields to describe the symmetry. SymmetryGAN (Desai et al., 2022) has also been used to detect rotational symmetry (Yang et al., 2023).

Another notable contribution to efforts to detect symmetries of data is *LieGAN*. LieGAN is a generative-adversarial network intended to return infinitesimal generators of the continuous symmetry group of a given dataset (Yang et al., 2023). LieGAN has been shown to detect continuous affine symmetries, including transformations from the Lorentz group. It has also been shown to identify discrete symmetries such as rotations by a fixed angle.

While most continuous symmetry detection methods attempt to discover symmetries which are affine transformations, the representation of infinitesimal generators using vector fields has led to the discovery of continuous symmetries which are not affine (Ko et al., 2024; Shaw et al., 2024). In one case, the domains of image data and partial differential equations are examined in particular (Ko et al., 2024).

Continuous symmetry detection is more difficult than discrete symmetry detection (Zhou et al., 2021) since the condition of invariance given as $f \circ S = f$ must hold for all values of the continuous

parameter of $S$. This is corroborated by the increasingly complex methods used to calculate even simple symmetries such as planar rotations (Benton et al., 2020; Dehmamy et al., 2021; Yang et al., 2023). Some methods introduce discretization, where multiple parameter values are chosen and evaluated. LieGAN does this by generating various transformations from the same infinitesimal generator (Yang et al., 2023). We believe a vector field approach, as has become more recently common (Otto et al., 2024), addresses the issue of discretization. A vector field-based method reduces the required model complexity of continuous symmetry detection while offering means to detect symmetries beyond affine transformations (Finzi et al., 2020; Shaw et al., 2024).

## 2.2 RELATED WORK: SYMMETRY ENFORCEMENT

Our work is most closely related to previous work on symmetry enforcement using vector fields (Finzi et al., 2020; Otto et al., 2024; Shaw et al., 2025), since we are directly adapting the analogous methods presented by the various authors for image data and common image transformations. This method is also analogous to Physics-Informed Neural Networks (PINNs) (Raissi et al., 2019). With PINNs, model training is regularized using differential constraints which represent the governing equations for a physical system. The method of symmetry enforcement employed here differs from PINNs since the differential constraints obtained using infinitesimal generators do not generally have the interpretation of defining governing equations for a physical system.

Continuous symmetry enforcement in images is far from new, and we note some recently-developed methods. Some methods seek to enforce symmetry by augmenting the training dataset according to known symmetries (Bergman, 2019). *Augerino* attempts to enforce symmetry using augmented data, though the symmetries are discovered from the data rather than given *a priori*. Another established method of enforcing symmetry is feature averaging, which is thought to be generally more effective than data augmentation (Lyle et al., 2020).

A growing number of methods seek to use symmetry to construct invariant or equivariant models without the need for augmented training data, and there is previous work accomplishing this using infinitesimal generators (Dehmamy et al., 2021; Yang et al., 2023; Finzi et al., 2020; Otto et al., 2024). Some previous work addresses specific cases: the special case of compact groups is studied (Bloem-Reddy & Teh, 2020), and the case of equivariant CNNs on Homegeneous spaces is studied (Cohen et al., 2019). Other work speaks to the universality of invariant architectures (Maron et al., 2019; Keriven & Peyré, 2019; Yarotsky, 2018).

## 2.3 DIAGONAL GROUP ACTIONS AND THEIR INFINITESIMAL GENERATORS

Fundamental to our approach is the concept of multiparameter groups and their infinitesimal generators. As these topics are discussed in existing literature Shaw et al. (2024; 2025), we provide an overview only in Appendices B and C. Of fundamental importance for the specialization to image data is the subject of infinitesimal generators of diagonal group actions, which we now discuss.

Let $G$ be a group acting on a set $Y$. The action of $G$ on $Y$ induces an action on the set $Y^m$, where a group element $g \in G$ acts on $(y_1, y_2, \ldots, y_m) \in Y^m$ by acting on each component $y_i$ separately and in the manner in which $g$ acts on $y \in Y$. It is said that the action of $G$ on $Y^m$ is a diagonal action.

If $\Psi(t, \mathbf{x})$ is a flow on $\mathbb{R}^n$, the induced diagonal action on $\mathbb{R}^n \times \mathbb{R}^n \times \cdots \times \mathbb{R}^n = (\mathbb{R}^n)^m$ is defined:

$$\tilde{\Psi}(t, (\mathbf{x}_1, \mathbf{x}_2, \ldots, \mathbf{x}_m)) = (\Psi(t, \mathbf{x}_1), \Psi(t, \mathbf{x}_2), \ldots, \Psi(t, \mathbf{x}_m)).$$

If $X$ is the vector field infinitesimal generator for $\Psi(t, \mathbf{x})$, we denote the infinitesimal generator for $\Psi(t, \mathbf{x}_i)$ by $X_i$, so that the infinitesimal generator $\tilde{X}$ for $\tilde{\Psi}$ is given as

$$\tilde{X} = X_1 + X_2 + \cdots + X_m.$$

For example, let $\Psi(t, (x, y)) = (x + t, ty)$, so that $X = \partial_x + y\partial_y$, the level curves for which can be written as $y = ce^x$. The infinitesimal generator for the diagonal action of $\Psi$ on $\mathbb{R}^2 \times \mathbb{R}^2 \times \mathbb{R}^2$ is

$$\tilde{X} = \partial_{x_1} + y_1\partial_{y_1} + \partial_{x_2} + y_2\partial_{y_2} + \partial_{x_3} + y_3\partial_{y_3}.$$

Our model assumption is that an image with $m$ channels is an element of $(\mathbb{R}^n)^m$, with $n$ being the number of pixels. This merely formalizes the concept of a group "acting on each image channel

separately." However, another fruitful perspective outside the context of image data deals with samples, where $m$ represents the number of data points in a given dataset and where $n$ represents the number of features. It is within this context that the sample mean can be said to be *equivariant* under certain affine transformations: a group acts on each point $\{\mathbf{x}_i\}_{i=1}^m$ separately, where $\mathbf{x}_i \in \mathbb{R}^n$, so that the sample mean of $\tilde{\Psi}(t, (\mathbf{x}_1, \mathbf{x}_2, \ldots, \mathbf{x}_m))$ coincides with $\Psi(t, \cdot)$ applied to the sample mean of $\{\mathbf{x}_i\}_{i=1}^m$.[1]

## 3 METHODS

We model grayscale images as matrices: an image $I$ is in $\mathbb{R}^{p \times q}$, where $p \times q$ is the number of pixels. We model multi-channel images as tensors–that is, matrices of size $p \times q$ for each channel. A diagonal action for an RBG image thus admits multiple interpretations, as it can be (1) a diagonal action on $p \times q$ copies of $\mathbb{R}^3$, or (2) a diagonal action on three copies of $\mathbb{R}^{p \times q}$.

Another common model of group actions on images involves modeling images as sections of a vector bundle on $\mathbb{R}^2$: that is, as mappings $I : \mathbb{R}^2 \to \mathbb{R}^c$ (for a $c$-channel image). Rotations and translations, among other planar transformations, are naturally considered by defining group actions in the plane, and this is the more common approach to symmetry detection and enforcement for image data. In this paper, however, we do not consider group actions on the base space, separating our work from that of previous work.

### 3.1 FRAMEWORK FOR CONTINUOUS SYMMETRY DISCOVERY

Previous work has dealt with the discovery of vector field infinitesimal generators (Shaw et al., 2024; 2025). The approach previously employed can be summarized as the following multi-step framework:

1. Utilize a parametric assumption about the form of the generators $\{X_j\}$ of the multi-parameter group action
2. Obtain a smooth function $f$ to examine for symmetry
3. Optimize the parameters of the generator(s) sought using $X_j(f) = 0$

For image data, using our model of images as tensors, steps 1 and 2 require modification for two reasons: (1) image data are high-dimensional; (2) not all transformations respect the spatial structure of images.

### 3.1.1 ADAPTING PARAMETRIC ASSUMPTIONS FOR IMAGE DATA

The high-dimensional problem is no small matter. To detect affine symmetry for the MNIST (Deng, 2012) dataset, for example, one would require at least $28^4 > 600,000$ parameters to be present in the parametric form of the generator, which problem is duplicated for matrix representations, the sizes of which would be $(28 \times 28) \times (28 \times 28)$. For context, this is an order of magnitude larger than the number of training samples available, and closer to two orders of magnitude larger than the number of parameters needed to train a successful classifier on the test dataset. The problem is much worse for ImageNet (Deng et al., 2009), where the sizes of the images are $224 \times 224$: generators with over 2.5 **billion** parameters would be needed, not even counting the need to duplicate across three separate channels. And as is the case for MNIST, this large number of parameters is at least one order of magnitude (though more often three orders of magnitude) larger than the maximum of the number of training images and the number of parameters needed to train a classifier with reasonable performance.

To address this issue, we make one or more parametric assumption(s) about the generators which (1) involve exponentially fewer parameters than described above; (2) are likely appropriate specifically for image data. The first of these assumptions is that the unknown group action acts diagonally: either the same group action on $\mathbb{R}^c$ at each of the $p \times q$ pixels or else the same group action on $\mathbb{R}^{p \times q}$

---

[1]We note that equivariance under flows, like invariance, can be characterized in terms of the Lie derivative (Otto et al., 2024), though the subject of the current work is invariance and not equivariance.

across each $c$ channels. In the first case, symmetry discovery takes place in $\mathbb{R}^c$ rather than $\mathbb{R}^{c \times (p \times q)}$: note that $p \times q >> c$, generally. In the second case, symmetry discovery takes place in $\mathbb{R}^{p \times q}$ instead of $\mathbb{R}^{c \times (p \times q)}$: a reduction only of a small amount. The first case is appropriate for many types of images transformations, including the power law transform as well as adjusting brightness and hue. The second case is appropriate for transformations of pixels which depend on the values of other (perhaps neighboring) pixels, which takes us to our second restrictive assumption.

The second useful restrictive assumption is that the group action can be represented as a convolution of a fixed kernel applied to each image. This can leverage the second case of diagonal actions in the case where a kernel matrix defines a convolution across each channel separately. But more importantly for the reduction of the number of parameters, an affine transformation represented as a convolution of a kernel with images a parameter count equal to the size of the kernel: for the Gaussian blur transformation, for example, the kernel is typically on the order of $7 \times 7$ or $15 \times 15$, a sharp reduction in the number of parameters when compared with the entire affine space for images. Additionally, a restriction of this type is appropriate for image data, owing to transformations and architectures which make explicit use of convolutions: in particular, they make use of the local (spatial) image structure.

### 3.1.2 ADAPTING THE LEARNING OF THE MANIFOLD STRUCTURE

The high-dimensional problem apparent in image data also requires the adaptation of the second step in the symmetry discovery framework. Previous work on symmetry discovery using vector fields points to the need to explicitly identify a smooth function to test for symmetry (Shaw et al., 2024): therein, the authors introduce *level set estimation*, in addition to making use of probability distributions. However, the method of level set estimation does not appear to be developed, as yet, for high-dimensional data, as the experiments conducted using it are restricted to 10 or fewer dimensions, which is far below what is required for images. Probability density estimation is also difficult in high dimensions: while density and level set estimation work with this framework in principle, we offer an adaptation to this step which may be better suited in higher dimensions.

Let $I$ be an image in the training set, and let $\Psi(t, \cdot)$ be a flow. For each $I$, we assume an ordered sequence of transformed images $\{\Psi(t_j, I)\}_{j=1}^k$ (with $t_j < t_{j+1}$) and that there is an index $1 \leq i \leq k$ with $\Phi(t_i, I) = I$: this is a somewhat restrictive setting, though not unlike trajectory-based symmetry discovery that appears elsewhere (Yang et al., 2023). The vector field infinitesimal generator for the flow at $I$ is found by computing $\frac{d}{dt}\left(\Phi(t, I)\right)|_{t=t_i}$, which can be approximated using numerical methods such as finite differences.

### 3.2 FRAMEWORK FOR CONTINUOUS SYMMETRY ENFORCEMENT

As with continuous symmetry discovery, symmetry enforcement using vector field regularization has been previously proposed (Shaw et al., 2024; 2025), the steps of which can be given as the following multi-step framework:

1. Identify the infinitesimal generators $\{X_j\}$ corresponding to the symmetry to be enforced
2. Regularize the training of a smooth model $f$ using $X_j(f) = 0$ and an estimated gradient of $f$ (with respect to the inputs, not parameters)

The only adaptation of the symmetry enforcement framework needed is that which naturally follows our adaptation of the discovery framework: the transformations and, subsequently, the vector field infinitesimal generators, are specified using diagonal group actions, convolutions, or a combination of the two.

### 3.3 THE PRACTICALITY OF A VECTOR FIELD APPROACH FOR IMAGES

In image classification tasks, it is common practice to augment using several types of transformations. Thus, it is natural to consider symmetry regularization for multi-parameter groups, the infinitesimal generators of which were discussed in Section C. In practice, multi-parameter groups could be handled in multiple ways. Our approach is to "stack" the tensors defined by applying the components of the infinitesimal generators to the data, so that the regularization term has a larger shape than in the single-parameter case.

In general, flows of vector fields do not commute. As a simple example, planar rotations about the origin do not commute with translations in the horizontal direction. In fact, this can be seen by analyzing the Lie bracket of two vector fields: the flows of the vector fields commute if and only if the Lie bracket of the vector field vanishes identically. We see this in the simple example: the Lie bracket of $\partial_x$ and $-y\partial_x + x\partial_y$ is $\partial_y$. As our experiments rely on the power law transform and Gaussian blur, it should also be said that these transformations do not commute.

The presence of non-commuting transformations presents an additional consideration for data augmentation: a collection of transformations could be composed in any possible order, if the transformations do not commute. When considering the set of possible augmentations, this number grows rather quickly. Without considering order, $m$ augmentation types (transformations) with $n$ selected hyperparameter values each (discretizing $m$ continuous transformations) would yield a hyperparameter grid of size $n^m$. However, when considering the order in which these augmentations take place, the number of possible ways to uniquely augment a sample increases much faster: as there are $m!$ ways to order the transformations, we have $m! \cdot n^m$ unique augmentations. Even for small values of $n$, this increases combinatorially. However, this does not, in principle, affect the regularization approach, since we simply require that $X_i(F) = 0$ for each generator $X_i$.

There is also another practical consideration that leads us to consider a vector field approach. We will see in the experiments below that although symmetry regularization can obtain a similar effect as augmentation, augmentation has a tendency to produce models which are more symmetric, and not always at the expense of accuracy. The computational overhead of computing the model gradient for symmetry regularization, despite the fact that this gradient can be used for multiple symmetry terms in multi-parameter enforcement, is also a practical concern, leaving one to wonder whether symmetry regularization has practical value. One reason symmetry regularization is of practical value is due to the fact that the symmetry group, be it single- or multi-parameter, could be unknown before performing symmetry enforcement in a particular classification task. In the case where vector fields are used to characterize the symmetry, direct augmentation is generally difficult, owing to the need to compute the flow of a vector field (or many), which requires solving a system of first-order ordinary differential equations. Symmetry enforcement through vector field regularization, on the other hand, eliminates the need to identify the symmetry group explicitly in order to enforce the symmetry. Thus, a critical practical use of vector field regularization is that it enables the use of discovered continuous symmetries described by vector fields, which method of discovery is both efficient and expressive (Shaw et al., 2024).

The need to identify a function to test for symmetry may seem to restrict the ability to perform symmetry discovery in the context of image data. However, this need is also what allows symmetry discovery of models themselves to take place. A successfully-trained model may be probed for its symmetry properties, which symmetries may subsequently be used to regularize the training of other models. This approach is, of course, not unique to the case of images data: however, the restrictive assumptions on the vector fields themselves (diagonal actions and convolutions) allow for symmetry discovery of models to be completed without undue computational resources (we experiment with this in 4.3).

## 4 EXPERIMENTAL RESULTS

Our first group of experiments deals with symmetry enforcement. Our first two experiments showcase symmetry enforcement for two types of transformations, namely the power law transform and Gaussian blur. Our third experiment tackles symmetry enforcement in the case of a subset of Imagenet using a pretrained ResNet50 model (He et al., 2015; Deng et al., 2009). After symmetry enforcement experiments, we perform a symmetry discovery experiment, simulating the case where symmetry is discoverable in data but where the symmetry group is not explicitly recovered.

### 4.1 SYMMETRY ENFORCEMENT RESULTS

In this paper, we concern ourselves primarily with the power law transformation and Gaussian blur. The application of the power law transformation is sometimes called "gamma correction," as fractional parameter values lead to an image corrected for darkness.

### 4.1.1 POWER LAW SYMMETRY ON MNIST

In this section, we consider single-parameter symmetry enforcement experiments which make use of the power law transform. The infinitesimal generator for this transformation is derived in Appendix A.1. We train a simple convolutional neural network using the MNIST dataset in four distinct approaches. Each approach can be described in terms of Equation 20. Each approach uses the cross entropy loss for $\mathcal{L}$, the *Adam* optimizer (Kingma & Ba, 2014) with a learning rate of 0.001, and trains for twelve epochs. The Baseline approach uses $\lambda(t) = 0$. The Regularization approach uses $\lambda(t) = 0$ for the first three epochs, followed by $\lambda(t) = 0.5$ for the remaining epochs, where $\tilde{\mathcal{L}}$ is the mean square error loss scaled by $28^2$. The Augmentation approach is similar to the Baseline approach, but applies the power law transform according to a random (uniform) sample of $\gamma$-values from $\{0.25, 1.0, 1.75\}$. Table 1 summarizes our results for 10 independent trials (median $\pm$ IQR/2) for select values of $\gamma$. Values through 1.9 were not substantially different from values at 1.0 and are thus not shown.

Table 1: Power Law Transformation Results for the MNIST dataset (Median $\pm$ IQR/2)

| $\gamma$ | Baseline | Regularization | Augmentation | Reg + Aug |
|---|---|---|---|---|
| 0.1 | $0.4435 \pm 0.0757$ | $0.7289 \pm 0.0528$ | $0.9643 \pm 0.0318$ | $0.9066 \pm 0.0438$ |
| 0.2 | $0.8670 \pm 0.0332$ | $0.9497 \pm 0.0365$ | $0.9757 \pm 0.0315$ | $0.9275 \pm 0.0446$ |
| 0.25 | $0.9309 \pm 0.0359$ | $0.9604 \pm 0.0357$ | $0.9765 \pm 0.0317$ | $0.9295 \pm 0.0446$ |
| 1.0 | $0.9684 \pm 0.0413$ | $0.9690 \pm 0.0355$ | $0.9773 \pm 0.0323$ | $0.9317 \pm 0.0449$ |

Table 1 shows that Regularization can encourage a model to be more symmetric than a baseline approach while still maintaining fairly high accuracy on the original test set. However, the Augmentation approach appears to outperform Regularization for $\gamma = 0.1$. Curiously, a combined Regularization/Augmentation approach seemed to underperform Augmentation and Regularization, generally.

### 4.1.2 GAUSSIAN BLUR SYMMETRY ON MNIST

The experimental setup is similar to that of Section 4.1.1. The primary difference is that the infinitesimal generator is that of the (estimated) generator for Gaussian blur, derived in Appendix A.2. Additionally, the symmetry loss $\tilde{\mathcal{L}}$ is scaled by $10 \cdot 28^2$ rather than $28^2$, and the models are trained for 10, not 12, epochs.

Table 2: Gaussian Blur Transformation Results for the MNIST dataset (Median $\pm$ IQR/2)

| $\sigma$ | Baseline | Regularization | Augmentation | Reg + Aug |
|---|---|---|---|---|
| 0.1 | $0.8924 \pm 0.0024$ | $0.9710 \pm 0.0439$ | $0.9216 \pm 0.0409$ | $0.9797 \pm 0.0022$ |
| 1.0 | $0.8867 \pm 0.0027$ | $0.9594 \pm 0.0421$ | $0.9225 \pm 0.0405$ | $0.9714 \pm 0.0034$ |
| 2.0 | $0.8451 \pm 0.0138$ | $0.8731 \pm 0.0438$ | $0.9167 \pm 0.0391$ | $0.8969 \pm 0.0022$ |
| 3.0 | $0.7747 \pm 0.0201$ | $0.7524 \pm 0.0618$ | $0.9103 \pm 0.0371$ | $0.8108 \pm 0.0156$ |
| 6.0 | $0.7076 \pm 0.0354$ | $0.6288 \pm 0.0389$ | $0.8998 \pm 0.0346$ | $0.7234 \pm 0.0485$ |

The results of Table 2 demonstrate that the regularization parameter $\lambda(t)$ can be chosen such that the model performance on the original test is much more favorable than for transformed copies.

### 4.1.3 RESNET50 ON IMAGENETTE

To examine the ability of symmetry regularization to apply to large models, we look to the ResNet50 model (He et al., 2015) on a 10-class subset of ImageNet (Deng et al., 2009) known as ImageNette (Paszke et al., 2019). We do not adjust the model architecture, but rather map the predicted index to an index between 0 and 9 to obtain an adjusted prediction.

We train the model for two epochs without any pretrained weights. With regard to Equation (20), we select the Cross Entropy loss function for $\mathcal{L}$, $224 \cdot 224 \cdot 3$ multiplied by the Mean Square Error

for $\tilde{\mathcal{L}}$, and $\lambda(t) = 0.5$. We optimize using SGD (Sutskever et al., 2013) with momentum 0.9 and learning rate 0.01. We repeat this process five times, evaluating both on the original validation set as well as a blurred copy of the validation set with $\sigma = 3.0$ and a kernel size of 15. (For regularization, the infinitesimal generator is derived in a similar manner as in Appendix A.2, but with a kernel size of 15 instead of seven.) Subsequently, we repeat this process for a model using no regularization. No additional strategies to improve model training are employed.

The accuracy score of the unregularized model in terms of the mean $\pm$ standard deviation is $39.02\% \pm 4.74\%$ for the original validation set and $27.7\% \pm 3.33\%$ evaluating at $\sigma = 3.0$. For the regularized model, the scores are $47.87 \pm 1.29$ and $33.49 \pm 2.47$, respectively. This result not only shows a significant improvement obtained using symmetry regularization, but also that symmetry regularization can be applied for larger problems.

## 4.2 Symmetry Discovery from Data

We now demonstrate the discovery of the infinitesimal (vector field) generator for Gaussian blur symmetry. As discussed in Section 3.3, the symmetry group, be it single- or multi-parameter, could be unknown before performing symmetry enforcement in a particular classification task, creating a need to discover the symmetry. In this experiment, we simulate an unknown symmetry group present in the MNIST dataset by first augmenting the dataset according to $\sigma \in \{0.1i\}_{i=0}^{10}$. Next, for each original training image and its augmentations, we compute the gradient along $\sigma$ for each $\sigma \in \{0.1i\}_{i=0}^{10}$.[2] Generally, the infinitesimal generator is found by evaluating the derivative at the group identity–in this case, zero. However, as is explained in Appendix A.2, we approximate the infinitesimal generator by evaluating at $\sigma = 0.3$.

We have thus estimated a gradient image for each training image, which gradient image, up to scaling, should correspond with the output of the Gaussian blur infinitesimal generator applied to the training image. As we represent the Gaussian blur infinitesimal generator in Appendix A.2 by convolving an image with a 7x7 kernel, we can estimate this kernel by training a single-layer convolutional neural network (with a kernel of size 7x7), using the training images as input and the gradient images as targets. We train this small model using the L1Loss function and the Adam optimizer (Kingma & Ba, 2014), with a learning rate of 0.01, for 50 epochs. We obtain the following (approximate) 7x7 kernel, which has a cosine similarity of 0.9998 with respect to the estimate ground truth kernel:

$$\begin{bmatrix} 4.35 \cdot 10^{-3} & 2.35 \cdot 10^{-3} & 1.01 \cdot 10^{-3} & 4.90 \cdot 10^{-4} & -8.28 \cdot 10^{-5} & 6.33 \cdot 10^{-5} & 4.38 \cdot 10^{-4} \\ 2.95 \cdot 10^{-3} & 1.48 \cdot 10^{-3} & 1.90 \cdot 10^{-3} & 5.68 \cdot 10^{-3} & 6.30 \cdot 10^{-4} & 5.45 \cdot 10^{-5} & 4.42 \cdot 10^{-4} \\ 2.07 \cdot 10^{-3} & 1.59 \cdot 10^{-3} & 2.08 & 47.3 & 2.08 & -1.04 \cdot 10^{-5} & 3.70 \cdot 10^{-4} \\ 1.30 \cdot 10^{-3} & 5.01 \cdot 10^{-3} & 47.3 & -198 & 47.3 & 4.17 \cdot 10^{-3} & 5.47 \cdot 10^{-4} \\ 1.06 \cdot 10^{-3} & 1.26 \cdot 10^{-3} & 2.08 & 47.3 & 2.08 & 8.22 \cdot 10^{-4} & 6.92 \cdot 10^{-4} \\ 1.51 \cdot 10^{-3} & 1.33 \cdot 10^{-3} & 1.71 \cdot 10^{-3} & 5.24 \cdot 10^{-3} & 1.08 \cdot 10^{-3} & 3.94 \cdot 10^{-4} & 4.82 \cdot 10^{-4} \\ 2.14 \cdot 10^{-3} & 1.60 \cdot 10^{-3} & 1.27 \cdot 10^{-3} & 1.25 \cdot 10^{-3} & 8.67 \cdot 10^{-4} & 3.20 \cdot 10^{-4} & 5.86 \cdot 10^{-5} \end{bmatrix}$$

## 4.3 Model Symmetry Discovery

In this experiment, we consider the problem of symmetry discovery of a pretrained model. We begin by training a small CNN $F$ on the MNIST dataset, and we note that the accuracy on the test set is approximately 0.9716. We now seek to approximate a generator $X$ such that $X(F) = 0$ for each training image. We assume that the coefficient functions of $X$ are expressible in terms of a $3 \times 3$ convolutional kernel: that is, there is a $3 \times 3$ matrix $K$ such that $X(F)$ evaluated at an image $I$, denoted $X(F)|_I$, is given as

$$X(F)|_I = K * \nabla F(I).$$

Thus, with $F$ fixed, we seek to learn $K$ such that

$$K * \nabla F(I) = 0, \qquad ||K|| = 1. \tag{1}$$

---

[2] Previous work has computed vector field generators from level sets (Shaw et al., 2024). However, due to the high dimensionality of image data, a directional derivative along the flow path may yield a more efficient estimation algorithm.

We impose the constraint $||K|| = 1$ so that the learned weights of $K$ do not become identically 0: we reiterate that our discovered $K$ is equivalent to any uniform scaling of $K$.

Our symmetry discovery task is thus one of training a CNN with a single convolutional layer defined by a $3 \times 3$ kernel (using a stride of one, and where each input image is padded with zeros so that the shape of the output image is that of the input image). The constraint $||K|| = 1$ is applied manually by dividing the kernel, during each epoch, by its own magnitude. We use the L1 loss function (scaled by $10 \cdot 28^2$), optimizing using the Adam(Kingma & Ba, 2014) algorithm with a learning rate of $0.01$, training for 100 epochs. The result is the following kernel, which we take to define the discovered symmetry:

$$\begin{bmatrix} -0.5993 & 0.9755 & -0.5215 \\ 0.9917 & -1.7328 & 1.0194 \\ -0.4998 & 0.9674 & -0.6031 \end{bmatrix}.$$

## 5    CONCLUSION

In this paper, we have introduced continuous symmetry discovery and enforcement for image data using the Lie derivative along vector fields. We have shown that symmetry regularization can yield similar outcomes to augmentation. We have also shown that symmetry can be discovered from image data directly, and while the efficient method of discovering symmetry using vector fields does not yield the symmetry group explicitly, the discovered vector field infinitesimal generator(s) can still be used to enforce symmetry in downstream tasks. Therefore, symmetry regularization allows one to leverage symmetry discovery using vector fields, which is computationally efficient when compared with existing methods (Shaw et al., 2024; Hu et al., 2025). While symmetry enforcement using augmentation requires direct knowledge or discovery of the symmetry group itself, the symmetry group for a Lie algebra of vector fields is, by assumption, the Lie group generated by the flows of a Lie algebra basis for the vector fields.

Future work includes optimizing the runtime of symmetry regularization. Due to the need to compute the gradient with respect to the input images, symmetry regularization, in our experiments, takes significantly more time than training without. It may be possible to reduce the computation time with a more efficient implementation (such as computing gradients across the output dimensions in parallel), and it may also be possible to used cached gradients from the layers of the parameter gradients obtained using backpropagation.

Despite the remaining future computational improvements to be made on symmetry enforcement, we reiterate the discussion in Section C with regard to computational challenges of augmentation. For $m$ transformation types in which $n$ hyperparameter values are selected for each, we showed that there are $m! \cdot n^m$ unique augmentations if the transformations do not commute (which is the general case). Thus, an efficient augmentation strategy of selecting random augmentations for training runs the risk of falling hopelessly short of the number of needed augmentations, while a more robust method (explicitly expanding the size of the dataset with augmentations) faces the daunting task of training on substantially larger datasets. Meanwhile, though the input gradient is, as implemented, computationally expensive, each vector field makes use of this gradient during enforcement: it need only be computed once for all of the vector fields. Therefore, an additional advantage of symmetry regularization is that it scales better as the dimension of the symmetry Lie group is increased.

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

# A  DERIVATION OF INFINITESIMAL GENERATORS

Here, we provide a detailed derivation of the infinitesimal generators used in our experiments. Section A.1 provides a derivation for Power Law symmetry, and Section A.2 does so for Gaussian Blur.

## A.1  THE POWER LAW TRANSFORM

The power law transformation, sometimes known as "gamma correction", is a transformation in which each channel value at each pixel–which value we denote as $I$–is adjusted according to the following equation:

$$I \rightarrow c \cdot I^{\gamma}, \tag{2}$$

where $c$ is a (non-zero) hyperparameter Paszke et al. (2019), and where $I \in (0, 1]$. For this group action, the group identity element is $\gamma = 1$: differentiating and setting $\gamma = 1$ yields an infinitesimal generator at each pixel of

$$X = cI \ln(I)\partial_I. \tag{3}$$

Since $cX(f) = 0$ if and only if $X(f) = 0$, an infinitesimal generator for the Power Law transformation acting diagonally on the RGB channels can be taken to be

$$X = R \ln(R)\,\partial_R + G \ln(G)\,\partial_G + B \ln(B)\,\partial_B.$$

Now let $(R_i, G_i, B_i)$ denote the RGB values at the pixel $i$, and suppose that each image in a dataset has $m$ pixels. The infinitesimal generator for the diagonal action of the power law transformation in $(R, G, B)$ coordinates is given as

$$\tilde{X} = \sum_{i=1}^{m} R_i \ln\left(\frac{R_i}{255}\right) \partial_{R_i} + G_i \ln\left(\frac{G_i}{255}\right) \partial_{G_i} + B_i \ln\left(\frac{B_i}{255}\right) \partial_{B_i}. \tag{4}$$

## A.2  GAUSSIAN BLUR

The Gaussian blur transformation is a transformation in which each channel value at each pixel $(i, j)$, denoted by $I_{ij}$, is replaced by a weighted linear combination of its former value and the values of its neighbors. The weights of this linear combination are determined by a Gaussian function. More concretely, consider the following function:

$$f(\sigma; x, y) = \frac{1}{2\pi\sigma^2}\exp\left\{-\frac{x^2 + y^2}{2\sigma^2}\right\}. \tag{5}$$

Let $k$ be the pre-determined size of the kernel, and let $q$ be the integer quotient of $k$ and 2. Then

$$I_{ij} \rightarrow \sum_{s=i-q}^{i+q} \sum_{r=j-q}^{j+q} w_{sr} I_{sr}, \tag{6}$$

where

$$w_{ij} = \frac{f(\sigma; i, j)}{\sum_{s=i-q}^{i+q} \sum_{r=j-q}^{j+q} f(\sigma; s, r)}.$$

This can be represented as a convolution of the image with pixel values $I_{ij}$ with the kernel with weights $w_{ij}$.

At first, one would wish to compute a vector field for this transformation parameterized by $\sigma$, where $\sigma = 0$ corresponds to the identity transformation. However, the derivative of $w_{sr}$ at $\sigma = 0$ is 0 identically: therefore, we approximate the derivative at 0 by evaluating the derivative at a nearby point, namely $\sigma = 0.3$. At this point, $w_{00} \approx 0.9847$, while $w'_{00} \neq 0$. For concreteness, the matrix of values this yields is given below:

$$
\begin{bmatrix}
2.46 \cdot 10^{-41} & 2.05 \cdot 10^{-29} & 2.73 \cdot 10^{-22} & 6.35 \cdot 10^{-20} & 2.73 \cdot 10^{-22} & 2.05 \cdot 10^{-29} & 2.46 \cdot 10^{-41} \\
2.05 \cdot 10^{-29} & 1.45 \cdot 10^{-17} & 1.57 \cdot 10^{-10} & 3.25 \cdot 10^{-8} & 1.57 \cdot 10^{-10} & 1.45 \cdot 10^{-17} & 2.05 \cdot 10^{-29} \\
2.73 \cdot 10^{-22} & 1.57 \cdot 10^{-10} & 1.08 \cdot 10^{-3} & 0.14 & 1.08 \cdot 10^{-3} & 1.57 \cdot 10^{-10} & 2.73 \cdot 10^{-22} \\
6.35 \cdot 10^{-20} & 3.25 \cdot 10^{-8} & 0.14 & -0.56 & 0.14 & 3.25 \cdot 10^{-8} & 6.35 \cdot 10^{-20} \\
2.73 \cdot 10^{-22} & 1.57 \cdot 10^{-10} & 1.08 \cdot 10^{-3} & 0.14 & 1.08 \cdot 10^{-3} & 1.57 \cdot 10^{-10} & 2.73 \cdot 10^{-22} \\
2.05 \cdot 10^{-29} & 1.45 \cdot 10^{-17} & 1.57 \cdot 10^{-10} & 3.25 \cdot 10^{-8} & 1.57 \cdot 10^{-10} & 1.45 \cdot 10^{-17} & 2.05 \cdot 10^{-29} \\
2.46 \cdot 10^{-41} & 2.05 \cdot 10^{-29} & 2.73 \cdot 10^{-22} & 6.35 \cdot 10^{-20} & 2.73 \cdot 10^{-22} & 2.05 \cdot 10^{-29} & 2.46 \cdot 10^{-41}
\end{bmatrix}
$$

## B    VECTOR FIELDS AND FLOWS

We now provide some background on vector fields and their associated flows (1-parameter transformations). We refer the reader to literature on the subject for additional information (Lee, 2012). Suppose that $X$ is a smooth (tangent) vector field on $\mathbb{R}^n$:

$$X = \alpha^i \partial_{x^i} := \sum_{i=1}^{n} \alpha^i \partial_{x^i}, \tag{7}$$

where $\alpha^i : \mathbb{R}^n \to \mathbb{R}$ for $i \in [1, n]$, and where $\{x^i\}_{i=1}^{n}$ are coordinates on $\mathbb{R}^n$. $X$ assigns a tangent vector at each point and can also be viewed as a function on the set of smooth, real-valued functions. E.g. if $f : \mathbb{R}^n \to \mathbb{R}$ is smooth,

$$X(f) = \sum_{i=1}^{n} \alpha^i \frac{\partial f}{\partial x^i}. \tag{8}$$

For example, for $n = 2$, if $f(x, y) = xy$ and $X = y\partial_x$, then $X(f) = y^2$. $X$ is also a *derivation* on the set of smooth functions on $\mathbb{R}^n$: that is, for smooth functions $f_1, f_2 : \mathbb{R}^n \to \mathbb{R}$ and $a_1, a_2 \in \mathbb{R}$,

$$X(a_1 f_1 + a_2 f_2) = a_1 X(f_1) + a_2 X(f_2), \qquad X(f_1 f_2) = X(f_1) f_2 + f_1 X(f_2). \tag{9}$$

These properties are satisfied by derivatives. A flow on $\mathbb{R}^n$ is a smooth function $\Psi : \mathbb{R} \times \mathbb{R}^n \to \mathbb{R}^n$ which satisfies

$$\Psi(0, p) = p, \qquad \Psi(s, \Psi(t, p)) = \Psi(s + t, p) \tag{10}$$

for all $s, t \in \mathbb{R}$ and for all $p \in \mathbb{R}^n$. A flow is a 1-parameter group of transformations. An example of a flow $\Psi : \mathbb{R} \times \mathbb{R}^2 \to \mathbb{R}^2$ is

$$\Psi(t, (x, y)) = (x \cos(t) - y \sin(t), x \sin(t) + y \cos(t)), \tag{11}$$

with $t$ being the continuous parameter known as the flow parameter. This flow rotates a point $(x, y)$ about the origin by $t$ radians.

For a given flow $\Psi$, one may define a (unique) vector field $X$ as given in Equation 8, where each function $\alpha^i$ is defined as

$$\alpha^i = \left( \frac{\partial \Psi}{\partial t} \right) \Big|_{t=0}. \tag{12}$$

Such a vector field is called the infinitesimal generator of the flow $\Psi$. For example, the infinitesimal generator of the flow given in Equation 11 is $-y\partial_x + x\partial_y$.

Conversely, given a vector field $X$ as in Equation 8, one may define a corresponding flow as follows. Consider the following system of differential equations:

$$\frac{dx^i}{dt} = \alpha^i, \qquad x^i(0) = x_0^i. \tag{13}$$

Suppose that a solution $\mathbf{x}(t)$ to Equation 13 exists for all $t \in \mathbb{R}$ and for all initial conditions $\mathbf{x}_0 \in \mathbb{R}^n$. Then the function $\Psi : \mathbb{R} \times \mathbb{R}^n \to \mathbb{R}^n$ given by

$$\Psi(t, \mathbf{x}_0) = \mathbf{x}(t) \tag{14}$$

is a flow. The infinitesimal generator corresponding to $\Psi$ is $X$. For example, to calculate the flow of $-y\partial_x + x\partial_y$, we solve

$$\dot{x} = -y, \quad \dot{y} = x, \qquad x(0) = x_0, \quad y(0) = y_0 \tag{15}$$

and obtain the flow $\Psi(t, (x_0, y_0))$ defined by Equation 11. It is generally easier to obtain the infinitesimal generator of a flow than to obtain the flow of an infinitesimal generator.

A smooth function $f : \mathbb{R}^n \to \mathbb{R}$ is said to be $X$-invariant if $X(f) = 0$ identically for a smooth vector field $X$. The function $f$ is $\Psi$-invariant if, for all $t \in \mathbb{R}$, $f = f(\Psi(t, \cdot))$ for a flow $\Psi$. If $X$ is the infinitesimal generator of $\Psi$, $f$ is $\Psi$-invariant if and only if $f$ is $X$-invariant.

## C  Infinitesimal Generators of Multi-Parameter Groups

Let $G \in \mathbb{R}^s$ be a group, and suppose $G$ acts on $\mathbb{R}^n$: that is, for $g_1, g_2 \in G$ and for $x \in \mathbb{R}^n$, there is a function $\Psi : G \times \mathbb{R}^n \to \mathbb{R}^n$ such that (assuming the group operation is vector addition)

$$\Psi(\mathbf{0}, x) = x, \qquad \Psi(g_2, \Psi(g_1, x)) = \Psi(g_1 + g_2, x). \tag{16}$$

The use of the symbol $\Psi$ to denote a multi-parameter group action is not accident, as a flow is a 1-parameter group action. Let $\{v_i\}_{i=1}^s$ be a basis for the tangent space of $G$ at $\mathbf{0}$, the group identity element. Lastly, let $\sigma$ be a curve in $G$ for which $\sigma(t_0) = \mathbf{0}$ and $\dot{\sigma}(t_0) = v_i$ for $t_0 \in \mathbb{R}$. The infinitesimal generator $X_i$ corresponding to $v_i$ is given by

$$X_i = \left( \frac{d}{dt} \Psi(\sigma(t), x) \right) \bigg|_{t=t_0}. \tag{17}$$

For example, consider the group $G = \mathbb{R}^3$ acting on $\mathbb{R}^2$ via

$$\Psi((a, b, \theta), (x, y)) = (x \cos(\theta) - y \sin(\theta) + a, x \sin(\theta) + y \cos(\theta) + b).$$

Given the following three curves,

$$\sigma_a(t) = (t, 0, 0), \qquad \sigma_b(t) = (0, t, 0), \qquad \sigma_\theta(t) = (0, 0, t),$$

we find that

$$X_a = \frac{d}{dt}(t, 0)\,|_{t=0} = \partial_x, \qquad X_b = \frac{d}{dt}(0, t)\,|_{t=0} = \partial_y,$$

$$X_\theta = \frac{d}{dt}(x \cos(\theta) - y \sin(\theta), x \sin(\theta) + y \cos(\theta))\,|_{t=0} = -y \partial_x + x \partial_y.$$

For each of these vector fields, a corresponding flow can be computed, which flows we call $\Psi_a$, $\Psi_b$, and $\Psi_\theta$, respectively. In terms of the original parameters, these flows are given as

$$\Psi_a(a, (x, y)) = (x + a, y), \qquad \Psi_b(b, (x, y)) = (x, y + b),$$

$$\Psi_\theta = (x \cos(\theta) - y \sin(\theta), x \sin(\theta) + y \cos(\theta)).$$

While each of these flows are, individually, 1-parameter group actions, it is clear that the infinitesimal generators $X_a$, $X_b$, and $X_\theta$ are the infinitesimal generators for the multi-parameter group action given in Equation (16). Thus, discovering vector field infinitesimal generators which annihilate a fixed (smooth) function does not solely apply to 1-parameter groups.

# D USING VECTOR FIELDS FOR DISCOVERY AND ENFORCEMENT

## D.1 QUANTIFYING THE EXTENT TO WHICH A SMOOTH MODEL IS INVARIANT

The notion of similarity of vector fields has been previously discussed (Shaw et al., 2025), though we include this discussion herein since the proposed methodology is not likely known to the machine learning community at large. Given a metric tensor $g_{ij}$–usually assumed to be the standard Euclidean metric tensor–we define the angle between two vector fields $X$ and $\hat{X}$ by

$$\cos\left(\theta(X, \hat{X})\right) = \frac{1}{\int_\Omega d\mathcal{M}} \mathbb{E}\left[\frac{|\langle X, \hat{X}\rangle_g|}{||X||_g \cdot ||\hat{X}||_g}\right], \tag{18}$$

where $\langle X, \hat{X}\rangle_g = \sum_{i,j} f_i \hat{f}_j g_{ij}$, $||X||_g = \sqrt{\langle X, X\rangle_g}$, and where

$$\mathbb{E}\left[u(\mathbf{x})\right] = \int_\Omega u(\mathbf{x}) d\mathcal{M},$$

with the region $\Omega$ being defined by the range of a given dataset. Ordinarily, this is the full range of the dataset. This formula is a generalization of the formula given in (Shaw et al., 2024) in the case where the manifold and/or metric is not assumed to be Euclidean.

The notion of a "cosine similarity" between vector fields induces a method by which the extent to which a particular smooth function is invariant can be quantified in relation to other functions. For a fixed vector field $X$, a function $f$ is $X$-invariant if and only if $X$ is orthogonal to the gradient of $f$, which vector field we denote $X_f$. Thus, the extent to which $f$ is $X$-invariant can be quantified in terms of the cosine of the angle between $X$ and $X_f$, given in Equation (18): the closer to 0 this value is, the more $X$-invariant $f$ is.

## D.2 CONTINUOUS SYMMETRY ENFORCEMENT USING VECTOR FIELD REGULARIZATION

In this section, we summarize the previously-given method (Shaw et al., 2025; Otto et al., 2024; Finzi et al., 2020) of enforcing continuous symmetries using regularization induced by vector fields. We seek to extend these approaches to image data.

Suppose that, as in the case of supervised learning, one seeks to learn a function $F$ mapping data instances $x_i$ to targets $y_i$, where $x_i \in \mathbb{R}^d$ and $y_i \in \mathbb{R}^m$. Suppose also that the function $F$ is estimated by means of the minimization of $\mathcal{L}(F(\mathbf{x}), \mathbf{y})$ for a smooth loss function $\mathcal{L}$. The model function $F$ is invariant with respect to the infinitesimal generators $\{X_k\}_{k=1}^s$ precisely when, for each component $f_j$ of $F$,

$$X_k(f_j) = 0 \tag{19}$$

for $1 \le k \le s$ and $1 \le j \le m$. This can be used as a regularization term giving a loss function of:

$$(1 - \lambda(t))\mathcal{L}(F(\mathbf{x}), \mathbf{y}) + \lambda(t)\tilde{\mathcal{L}}(\mathbf{X}(F)(\mathbf{x}), \mathcal{O}), \tag{20}$$

where $\mathbf{X}(F)(\mathbf{x}) = (X_k(f_j))(x_i)$, $\mathcal{O}$ is an array of zeros and is of the same shape as $\mathbf{X}(F)(\mathbf{x})$, $\tilde{\mathcal{L}}$ is a smooth loss function, and $\lambda(t) \in [0, 1]$ is a (possibly time/epoch-dependent) symmetry regularization parameter.

