# OpenReview forum: "Continuous Symmetry Discovery and Enforcement for Image Data"
_ICLR.cc/2026/Conference — Submitted to ICLR 2026_

### Official Review · Reviewer_SbJc · 2025-10-20

**Soundness:** 2
**Presentation:** 1
**Contribution:** 1
**Rating:** 2
**Confidence:** 5

**Summary:**

The paper discusses several potential issues in discovering and enforcing continuous symmetries for image data, including algebraic nonclosure and other violations of algebraic constraints from the discovered infinitesimal generators, the exponential growth of the number of transformations required for data augmentation with respect to non-Abelian symmetry groups, etc. The experiments evaluate some relevant methods, such as regularization and data augmentation for the power law symmetry in MNIST, and a preliminary result of symmetry discovery by inspecting the gradient of a trained predictor and finding its orthogonal vector field.

**Strengths:**

The paper reviews and discusses some important related works in symmetry discovery and enforcement, particularly for image data. A detailed background section is provided for vector fields as infinitesimal generators of continuous symmetry, so readers with relatively little experience in this field can understand the subject easily.

**Weaknesses:**

This paper, in my opinion, is mostly a review of existing methods. The contributions, if any, are very unclear. To see this, the related work section spans up to four pages, and even in the methodology section, Sec 3.1 and 3.2 are still restatements from existing work. The **contribution** paragraph at the end of Sec 1 states that the paper "provides a mathematical framework for the extension of continuous symmetry discovery and enforcement for image data", but no such framework is clearly presented in the current paper. Also, the paper is titled "... for image data", but most of the contents do not specify what is special about the symmetry in image data. Also, there is no explanation why existing methods would not work on image data. In fact, a lot of related work, whether mentioned in this paper or not, already showed results of symmetry discovery on image datasets as parts of their experiments.

From the narrative of the paper, Sec 3.3 and 3.4 is supposed to introduce some new methods for symmetry enforcement and discovery. However, I regret not finding any valuable new insights. For multi-parameter symmetry groups, stacking the tensors from multiple generators is a standard and straightforward technique which is already used in past works [1, 2]. For data augmentation w.r.t non-Abelian groups, I agree that augmenting with parameters on a fixed grid can result in at most combinatorial and exponential sample complexity. However, a simple yet effective alternative approach would be to randomly sample the group parameters. Finally, Sec 3.4 is titled "symmetry discovery", but the content of the subsection focuses elsewhere and has not clearly described any method for symmetry discovery.

Apart from the previously mentioned ones, there are other important missing references in this paper. For symmetry discovery, LieGG [3] trains a predictor and solves the symmetry of the predictor algebraically; LaLiGAN [4] parameterizes vector field symmetry by a composition of an autoencoder and a linear action. These are closely related to the subject of this paper and should be discussed and possibly compared against in the experiments.

### References

[1] Symmetry-Informed Governing Equation Discovery. NeurIPS, 2024.

[2] Symmetry Discovery for Different Data Types. Neural Networks, 2025.

[3] Liegg: Studying Learned Lie Group Generators. NeurIPS, 2022.

[4] Latent Space Symmetry Discovery. ICML, 2024.

**Questions:**

none

---

> ### Author Response · Authors · 2025-11-21
> **Initial response**
>
> We appreciate the time the reviewer has spent with our work. However, it seems that the reviewer may not have understood the key component of our work. In particular, with regard to the provided summary of our work, we do not merely discuss potential issues with symmetry discovery and enforcement for image data, but rather we introduce a novel construct by which symmetry discovery and enforcement may take place.
>
> *Weaknesses*
>
> Our related work sections comprise 1.5 pages: our paper is not mostly a review of existing methods. We do, on the other hand, build upon the vector field discovery and enforcement paradigm introduced in (Shaw et al., 2024 \& 2025), devoting text to reviewing these methods, since these methods do not seem to be well-understood within the discipline of machine learning. As we have also said in response to reviewer V48H, we are happy to place this material in appendices to avoid the appearance that these concepts are new as of this work.
>
> The paper does explain why existing methods would not be suitable for image data, which explanation appears in the penultimate paragraph of the introduction. However, reviewer skepticism has prompted us to explain this problem in more detail, which additional detail is given in a general comment. In short, the adaptation to image-specific infinitesimal generators is needed because existing methods face the task of optimization in an impractically large search space (which search space becomes much larger when more expressive symmetries are sought). Our work considers transformations on the level of pixels, not rotations and transformations: we are aware of no other work that can discover and/or enforce such symmetries. In particular, we are aware of no such work which can generalize to symmetries that cannot be represented using matrices (such as the power law transform) or which do not explicitly augment data.
>
> The novelty of our work is in our dealing with image-specific symmetries. As we discuss in our general response, even the efficient method of vector field symmetry discovery faces the high-dimensional problem of affine symmetry discovery for images. Our novel contribution allows for the discovery of symmetry for images in a much more efficient way, since we adapt the way in which the vector fields are represented (convolutions, diagonal group actions). To our knowledge, there are no methods in existence which have discovered Gaussian blur symmetry (or other pixel-level transformations) at all, let alone by tuning so few parameters (in our case, 49). And, to our knowledge, there are no other methods in existence, excepting augmentation, which can enforce symmetries which are not affine for image data.
>
> Our own implementation of augmentation in the experiments is precisely what is described by the reviewer: randomly sampling augmentations. However, we are aware of no work which considers the notion that the order of transformations during augmentation may matter: it seems that ignoring this entirely is common practice. This may be a valid technique for certain common image transformations, particularly owing to the well-established success of augmentation. However, for image transformations in general, we hypothesize that the technique of random sampling may become less effective as the number of symmetry types increases: the random sample will only cover a very small amount of possible augmentations. But to our knowledge, this problem has not been explored at length, and we merely pose it as a potential problem, particularly for more complex symmetries.
>
> We do not wish for confusion regarding section 3.4. In this section, we have provided motivation for symmetry enforcement using vector field regularization: to capitalize on the efficiency gains of symmetry discovery. We can reorganize this segment in the revision, so that the method of symmetry discovery is given and our desired segment on the motivation appears elsewhere.
>
> We appreciate the reviewer bringing up other related work, and we can include these in the related work section. However, as discussed in the general response, such methods of symmetry discovery and enforcement are simply not practical for the image transformations we are considering: the symmetry search space is simply too large. We obtain a reduction of this search space under the diagonal group action assumption and/or the assumption that the vector field components can be represented using convolutions. Our experiments are completely beyond the reach of existing methods, to our knowledge.

---

> > ### Comment · Reviewer_SbJc · 2025-11-22
> >
> > I appreciate the authors' efforts in providing the response and the additional experiments. I would like to highlight my previous comment that **the paper does not clearly present a methodological framework**. In particular, to address symmetry discovery in image data, I believe the following components are necessary:
> > * **Define the group action explicitly wrt the image signal**. Suppose each image is a feature field $f: \mathbb R^2 \to \mathbb R^d$. Does the symmetry transform the base space $\mathbb R^2$ or the feature space $\mathbb R^d$? Or both? I understand your comment that some existing methods, like LieGAN and LieGG, can only handle base-space symmetries. But you should make this statement explicitly in the paper, rather than just saying that these methods do not work for (general) image symmetries.
> > * Following the previous point, if you consider **transformations on both the base space and the feature space**, like the Gaussian blur in your experiment, you need to justify **why they can be seen as the action of a symmetry group**. In fact, Gaussian blur is not considered a symmetry in the usual sense we talk about group invariance/equivariance in machine learning. The blurring transformation is not invertible and cannot be described in terms of a vector field as the generator of a one-parameter subgroup. Therefore, much of the result in Sec 4.2 does not make sense.
> > * **The objective of symmetry discovery**. Given a dataset, what does it mean that a certain set of transformations is the symmetry of the data? Is it the group invariance/equivariance of a function, or some other criterion? This objective is not defined in the current submission. The experiment in Sec 4.2 on symmetry discovery is also confusing. It seems that the model is trained to learn a localized kernel of Gaussian blurring from a synthetic dataset of pairs of original and blurred images. This setting seems more "supervised" and different from existing methods for symmetry discovery without such pair information. The authors should explain why this difference and why the setup used in this paper is valid and meaningful.
> >
> > I think the paper has the potential to improve with better writing and clearer presentation. However, at its current state, it would require significant modifications to address the concerns mentioned above and in my original review. Therefore, I will keep my original score.

---

> > > ### Author Response · Authors · 2025-11-29
> > >
> > > We thank the reviewer for additional helpful comments. We have revised the paper according to previous feedback and the current feedback. In short, we
> > >
> > > 1. moved the sections reviewing vector fields and symmetry enforcement to the appendices
> > >
> > > 2.  combined subsections of the previous "Experimental Methods" section under a new "Methods" section, and the previous material appears under "The Practicality of a Vector Field Approach for Images." The earlier sections of the Methods section outline the framework, together with our alterations, explicitly. This section also explicitly gives the group action for our mathematical model of images: note that the transformations for Gaussian blur and the power law are given explicitly in the appendices, where the infinitesimal generators are derived.
> > >
> > > 3. provided an illustrative experiment for the task of learning symmetries of a pretrained model (as per reviewer sigs' request to provide additional applications for our method).
> > >
> > > We will place the additional experiments we have conducted, viewable on this page, in an appendix called "Additional Experiments" for the camera-ready submission.
> > >
> > > This revision addresses the current feedback as follows.
> > >
> > > 1. We have specified that the group action takes place in $\mathbb{R}^{p \times q}$, not the base space. Other methods could attempt this: there is no theoretical reason they can't. The issue is entirely practical, owing to the high number of dimensions in which this group action takes place.
> > >
> > > 2. The Gaussian blur transformation does not take place in both spaces: it transforms only the pixel values, so it is not a base space transformation. The fact that the transformation can be represented as a convolution means that it leverages image structure, which is entirely appropriate for image data: this is one reason components of vector fields *should* be specialized for image data. But the leveraging of image structure (local neighborhoods of pixels) should not be confused with the notion of a group action in the plane itself.
> > >
> > > Regarding whether the Gaussian blur itself is well-posed for this method, we acknowledge that we have merely approximated it. The Gaussian blur transformation is approximated at $\sigma=0.3$ rather than at the true identity. Near $\sigma=0.3$, the transformation is invertible, and the approximated infinitesimal generator can be explicitly given, as we have done in the appendices. But it is true that this particular transformation cannot be used *exactly* with vector field regularization, owing to the vanishing of the "true" infinitesimal generator given by taking the limit as $\sigma \to 0$. (This may partially explain the performance of regularization near the original test set: only small values of $\sigma$ are improved.) Another potential approach for enforcing blur in image classification could be to use a transformation which employs both blur and sharpening. In this case, a parameter would be given for which negative values would correspond to blurring, positive to sharpening, and the identity transformation would correspond to a parameter value of 0.
> > >
> > > However, we reiterate that this paper is not attached to any particular transformation or infinitesimal generator: we merely use the power law transform and the (approximated) Gaussian blur transformations as illustrative examples for the framework.
> > >
> > > 3. While the revised paper speaks to some of this, we should emphasize the issue with supervision. Symmetry discovery is indeed inherently unsupervised. However, in (Shaw et al., 2024 & 2025), symmetry discovery using vector fields is accomplished by constrained least squares estimation on $X(f)=0$, where the targets are all identically 0: therefore, it is solved by means of constructing a supervised problem. In 4.2, we merely change the targets to be that of estimated tangent vectors, which are scale-equivalent to the ground truth, and which we obtained via direct computation from the data: we are solving an unsupervised problem by constructing a supervised problem.

---

### Official Review · Reviewer_sigs · 2025-10-28

**Soundness:** 3
**Presentation:** 2
**Contribution:** 2
**Rating:** 4
**Confidence:** 3

**Summary:**

The paper extends Lie‑derivative methods to image data by deriving infinitesimal generators for transformations and using them for　symmetry regularization and discovery. The framework covers non‑invertible semigroup transforms such as Gaussian blur and supports multi‑parameter, non‑commuting combinations. Experiments on MNIST and ImageNet show that regularization can improve robustness and sometimes approach or improve over unregularized baselines.

**Strengths:**

S1. The generator‑based formulation is interesting and applies even to non‑invertible semigroup transforms like Gaussian blur, widening applicability. It is also appealing that combinations of non‑commuting transformations can be handled efficiently through the multi‑parameter setup.

S2. The background and preliminaries are clearly written, making the paper accessible to readers without a deep prior in differential geometry or Lie theory.

**Weaknesses:**

W1. The empirical improvements are modest. In Table 1, augmentation outperforms regularization in all reported settings, and in Table 2 regularization beats augmentation in only two of five settings. Other metrics beyond accuracy, such as training efficiency or compute overhead, are not evaluated.

W2. In the more realistic ImageNet experiment (Section 4.1.3), results are not compared against an augmentation baseline, so the practical significance of the method is unclear.

W3. The method relies heavily on prior vector‑field symmetry work (e.g. Shaw et al., 2025); the paper’s distinct technical contribution appears concentrated in Sections 3.3 and 3.4 and may not be substantial enough as currently presented.

**Questions:**

Q1. In Table 1, why does Reg+Aug underperform Aug alone? A simple intuitive explanation would help.

Q2. Can replacing augmentation with regularization reduce the amount of labeled data needed to reach a target accuracy (sample efficiency)?

Q3. Do you have results when combining several non‑commuting transformations in the same run, rather than one at a time?

Q4. Section 4.2 uses synthetically applied transformations. Can you demonstrate discovery on real data where such transformations occur naturally?

Minor comments
- Line 247: “Section 2.4” appears to be a typo for “Section 3.3”.
- Section 3.4 is text‑only and difficult to follow; a pseudocode or algorithm box would improve clarity.

---

> ### Author Response · Authors · 2025-11-21
> **Initial response**
>
> *Weaknesses*
>
> 1. We are not trying to outperform augmentation. We compare with augmentation to illustrate that, similar to augmentation, greater model generalization and/or performance can be obtained using regularization. Our experiments suggest that vector field regularization is primarily useful when augmentation is not possible or else prohibitively expensive: it makes use of the efficiency gains of vector field discovery and does not require explicit knowledge of the associated transformations.
>
> 2. The purpose of this experiment is to demonstrate that regularization can lead to improved outcomes when compared with the vanilla approach--that these outcomes are not just on account of the small MNIST dataset. We are not trying to outperform augmentation, though we have reason to suspect that augmentation may outperform regularization, as with MNIST.
>
> 3. We regard previous work connecting symmetry to vector fields to be quite enabling indeed. In this work, we exploit the flexible nature of this representation to specialize to image-specific transformations in ways that are not considered in previous work. In a general response, we point to the current problem of trying to employ affine symmetry detection for images: the problem is simply too high-dimensional without such specialization. Without our contribution, symmetry discovery and enforcement for image data (for the types of transformations we consider) would simply not be feasible.
>
> *Questions*
>
> 1. Unfortunately, there is no known simple explanation. Although vector field regularization can be applied for any differentiable model, it is possible that it is only for certain model architectures that true invariance can be met. Our work is thus a gateway to the study of many important questions, none of which are accessible without our current work.
>
> 2. Our experiments currently do not investigate this possibility, but this is an interesting question we hope we can provide experimental insight on before the discussion period closes.
>
> 3. Please see one of our general comments for an experiment combining Gaussian blur and the power law transform.
>
> 4. The primary application of symmetry discovery for images may turn out to be for synthetically-transformed data. For example, consider the issue of creating a skintone-neutral image classifier for images of faces. One could use generative models to incrementally adjust the skin tone of many training images: however, completing this task for an entire database of images is likely to be expensive. Due to the efficiency of vector field symmetry discovery, it may be possible to learn the complex symmetry corresponding to such a transformation, which symmetry could then be used to regularize model training.

---

> > ### Comment · Reviewer_sigs · 2025-11-27
> >
> > Thank you very much for the response.
> >
> > My main concern is that the practical benefits of the proposed approach are unclear. I think data augmentation is usually a better solution than tweaking the loss because of its simplicity. So I'd like to know the benefit of the proposal. According to the authors' reply, it is "useful when augmentation is not possible or else prohibitively expensive". Could you numerically evaluate the usefulness?
> >
> > The new results were interesting, but again, I couldn't find much gain from the augmentation.
> >
> > Since my concern has not been resolved, I'll maintain my original score.

---

> > > ### Author Response · Authors · 2025-11-29
> > >
> > > We thank the reviewer for the additional feedback. In our revised paper, we have included an experiment with demonstrates a scenario in which augmentation is not well-posed: discovering symmetries of pretrained models.
> > >
> > > Symmetry discovery using vector fields is the discovery of symmetries of *functions*, which, in the case of data itself, requires that the data is summarized using smooth functions (or tangent vectors to trajectories, as we have previously shown). But models themselves can be tested for symmetry. Thus, a potential application of our work is to learn symmetries of a pretrained model, using the learned symmetries in the training of subsequent models via vector field regularization. This can be termed "symmetry emulation," where a model can be trained to emulate the symmetries of another model.
> > >
> > > We do not believe this task can be accomplished practically using augmentation, as it would require a brute-force estimation method for symmetry discovery. While this task is not unique to the case of image data, this task would not be practical (see previous comments) without the image-specific forms of infinitesimal generators we provide in this paper.

---

### Official Review · Reviewer_a4Zt · 2025-10-31

**Soundness:** 2
**Presentation:** 2
**Contribution:** 2
**Rating:** 4
**Confidence:** 4

**Summary:**

This paper presents a framework for discovering and enforcing continuous symmetries in image data using the Lie derivative along vector fields. The key contribution is extending existing vector field-based symmetry methods (previously limited to affine transformations or tabular data) to common image transformations like power law correction and Gaussian blur. The authors derive infinitesimal generators (vector fields) for these transformations and show that symmetry can be enforced via regularization without data augmentation. They demonstrate: (1) symmetry enforcement via regularization achieves comparable performance to augmentation on MNIST and ImageNette, (2) the learned vector field generators can be discovered from augmented data with high accuracy (0.9998 cosine similarity for Gaussian blur), and (3) the approach scales to large models (ResNet50). The framework relies on diagonal group actions—where transformations act on each image channel or sample independently—and uses the model's gradient with respect to input images.

**Strengths:**

Clear problem motivation: The paper articulates well why vector field-based symmetry enforcement is desirable for image data—avoiding explicit augmentation and enabling use of discovered symmetries.

Mathematical framework: The extension from general vector field methods to diagonal group actions (Section 2.5) provides theoretical justification for the approach, and is easy to follow and also understand. The infinitesimal generators for power law and Gaussian blur transformations is interesting.

Multi-parameter symmetry handling: Section 3.3 discusses practical considerations for multiple non-commuting transformations, noting that augmentation faces combinatorial explosion ($m!·n^m$ augmentations) while regularization scales linearly.

The ResNet50/ImageNette experiment (Section 4.1.3) demonstrates scalability beyond toy problems, showing 8.85% absolute accuracy improvement with regularization (47.87% vs 39.02%). The results demonstrate 0.9998 cosine similarity between learned and ground-truth Gaussian blur generator (Section 4.2) shows the discovery framework can work.

**Weaknesses:**

Incomplete theoretical development: Diagonal action assumption not justified or validated empirically. No analysis of approximation error when generators are estimated (Gaussian blur is an "estimate").

Dataset limitations: Only MNIST (28×28, 1-channel, simple) and ImageNette (10 classes). No CIFAR-10/100, no full ImageNet, no other computer vision benchmarks.

Regularization often underperforms augmentation: At extreme parameter values (Table 1: γ=0.1, Table 2: σ=6.0), regularization significantly worse than augmentation.

Discovery experiment limitations: Only demonstrates recovery of known transformation from explicitly augmented data. Only single transformation tested (Gaussian blur). No analysis of failure modes or limitations.

This only works for transformations with tractable infinitesimal generators. Unclear how to discover vs enforce symmetries. No analysis of memory requirements or GPU utilization.

Limited novelty: The contribution is primarily applying existing vector field regularization methods to specific image transformations. The discovery experiment only shows recovery of known transformations from augmented data, not true discovery.

**Questions:**

Diagonal action validation: Can you provide empirical evidence that the diagonal action assumption holds for your target transformations? Have you tested on transformations where channels are coupled (e.g., RGB to grayscale, color temperature shifts)?

Generator derivations: Can you provide the complete derivations for the power law and Gaussian blur infinitesimal generators? Why is the Gaussian blur generator an "estimate" rather than exact?

When does regularization fail?: In Table 2, regularization dramatically underperforms at σ=6.0 (62.88% vs 89.98% for augmentation). Can you characterize when/why regularization fails? Is there a theoretical or empirical criterion?

Theoretical guarantees: Under what conditions does minimizing the regularization loss (Eq. 14) guarantee that the model will be invariant to the transformation? Are there cases where regularization can fail even with perfect optimization?

Baseline comparisons: LieGAN, Augerino can be usefule additions as baselines and this is missing.

Beyond cosine similarity, how can you validate that discovered generators are correct? Visualize the flow they generate?

How does the method scale to higher resolution images (224×224×3 for ImageNet)?

---

> ### Author Response · Authors · 2025-11-21
> **Initial Response - Weaknesses**
>
> *Weaknesses*
>
> 1. If the diagonal action assumption is not clearly justified in the paper as written, let us provide this justification more explicitly. Many common image transformations act diagonally. For example, the power law transform applies the same transformation to each pixel: for MNIST, it is an action on $\mathbb{R} \times \mathbb{R} \times \dots \times \mathbb{R}$. Therefore, we consider the restriction to diagonal group actions because the assumption that a discoverable symmetry acts diagonally on the images is consistent with several common image transformations: it is one aspect of our specialization to image transformations.
>
> 2. Below are enforcement results for CIFAR-10 using Gaussian blur symmetry. Each model was trained for 2 epochs (5 repetitions each). This shows the baseline winning on the original test set, but quickly becoming overtaken by the other methods. Augmentation scores the best, followed by the hybrid approach, followed by regularization.
>
> | $\sigma$   | Regularization | Reg + Aug | Augmentation | Baseline |
> |---------------------------|---------------------|---------------------|---------------------|---------------------|
> | $0.1$ | $0.5136 \pm 0.0048$ | $0.5232 \pm 0.0053$ | $0.5408 \pm 0.0070$ | $0.6047 \pm 0.0161$|
> | $1.0$ | $0.4917 \pm 0.0048$ | $0.5170 \pm 0.0011$ | $0.5369 \pm 0.0018$ | $0.4346 \pm 0.0051$|
> | $2.0$ | $0.4486 \pm 0.0076$ | $0.5083 \pm 0.0041$ | $0.5230 \pm 0.0035$ | $0.3008 \pm 0.0029$|
> | $3.0$ | $0.4337 \pm 0.0088$ | $0.5054 \pm 0.0060$ | $0.5148 \pm 0.0043$ | $0.2862 \pm 0.0059$|
> | $6.0$ | $0.4270 \pm 0.0086$ | $0.5018 \pm 0.0074$ | $0.5063 \pm 0.0050$ | $0.2791 \pm 0.0060$|
>
> 3. We are not trying to outperform augmentation. We are comparing to augmentation primarily to show that, like augmentation, regularization can lead to better model generalization. The main use case of symmetry regularization is when augmentation either cannot be done or is too expensive to do extensively, particularly when the symmetries are discovered. Symmetry discovery using vector fields, unlike existing methods (see the general response), is feasible for images, provided our novel adaptation is used. However, as the symmetries are not characterized explicitly (one is left with only the generators), regularization is a method by which the discovered symmetries can be put to use in enforcement.
>
> 4. See (6) below for a discussion of discovering known symmetries. See the general response for an experiment using both symmetries.
>
> 5. We fear that the flexibility of using vector fields as infinitesimal generators has not been fully appreciated: the coefficient functions can be quite expressive--in fact, in (Shaw et al., 2024), universal approximators are used. The distinction between symmetry discovery and enforcement is that, for discovery, a smooth function $f$ is fixed, and vector fields $X_j$ as estimated such that $X_j(f)=0$: in the enforcement case, a collection $X_j$ of vector fields is fixed, and we seek a smooth function $f$ such that $X_j(f)=0$. We will ensure that this is made sufficiently clear in the revision.
>
> With regard to computational requirements, we note the large efficiency gains noted in the general response: we believe that our method is clearly unrivaled by any SOTA methods.
>
> 6. As mentioned in the response to reviewer V48H, our contribution is nuanced, but not limited. The large efficiency gains are no small matter: after all, discovering and leveraging symmetry is usually meant only to complement the main machine learning task (image classification, in our case). To our knowledge, no other methods in existence can deal with the discovery and enforcement of the transformations we consider in a practical manner without use of augmentation. Without the specialization for image transformations that we provide, the vector field approach introduced by (Shaw et al., 2024) would be an impractically large problem.
>
> The discovery of known symmetries is much more meaningful in this context than the discovery of unknown symmetries: it allows us to compare the discovered symmetry to ground truth, thereby offering insight into the capabilities of our method.

---

> ### Author Response · Authors · 2025-11-21
> **Initial Response - Questions**
>
> *Questions*
>
> 1. As mentioned above, the justification for diagonal group actions simply comes from the fact that many common image transformations act diagonally (same transformation per pixel). Thus, no empirical evidence is needed to justify this restriction. Even in cases where channels are coupled, we still have the alternative interpretation of the group acting on each pixel separately.
>
> 2. The reason the Gaussian blur generator is an estimate rather than exact is due to the fact that, strictly speaking, the straight-laced calculation of the (normalized) Gaussian blur transformation leads to a vector field with identically zero coefficients. This is because the derivative of the transformation with respect to the standard deviation parameter at zero is identically zero. Therefore, we evaluate the derivative at a standard deviation *near* zero: at $\sigma=0.3$, the $1-$dimensional Gaussian is above $0.95$ (near 1, indicating proximity to the identity transformation), but the derivative is not zero.
>
> 3. There are many possible explanations as to why regularization may fail. Although we point out that any smooth model may be regularized, it is possible that the success of regularization depends on the model architecture: it is quite likely that a model for which $X(f)=0$ simply isn't possible to obtain for an arbitrary model architecture. This naturally raises the question as to what model architectures are appropriate for a given (set of) vector field(s): this question, however, cannot be answered in short order, likely requiring years of extended investigation. Our paper is thus the gateway to a myriad of interesting and consequential questions, which questions can only be considered subsequent to our work.
>
> The tuning of the regularization parameter is also not inconsequential. For the Gaussian blur experiment, we can obtain different results using a different regularization parameter.
>
> 4. As our method of regularization can be used for any differentiable model, there can be no general theoretical guarantees. This resembles ``classic overfitting,'' where it is possible that $X(f)=0$ on the training data, but the model is only symmetric within the training data regime. Our experiments suggest, however, that improved model generalization for test data can be obtained.
>
> 5. Our work considers a different type of symmetry than Augerino: transformations on the pixels themselves rather than rotations, translations, and the like. To our knowledge, no other method in existence takes up the symmetries we do. And we believe to try would be catastrophic, as explained in the general response.
>
> 6. Actually, the entire point of vector field symmetry discovery and enforcement is to *not* recover the flow explicitly: this is a computationally expensive operation, in general. By making use of regularization, we avoid the need to recover the flow of discovered symmetries.
>
> 7. In the context of enforcement, the scaling issue is primarily due to the need to approximate the model gradient. In our implementation, we have done this using automatic differentiation, and at every epoch. This introduces computational overhead similar to backpropagation, increasing the time needed per epoch. However, as mentioned in the paper, multiple symmetries can make use of this gradient, helping the method to scale to multiple symmetries.

---

### Official Review · Reviewer_V48H · 2025-11-02

**Soundness:** 3
**Presentation:** 3
**Contribution:** 1
**Rating:** 2
**Confidence:** 4

**Summary:**

This paper adapts the flow regularization method for enforcing symmetries, introduced in Shaw et al 2024, to images. The idea is to enforce continuous symmetries by requiring that their infinitesimal action preserves the loss.
They adapt this idea to images by assuming a "diagonal action" which means it acts on each image channel separately, or in image data, say it acts on each image independently.

They test their regularizer on predefined flows, such as gamma correction and gaussian blur on MNIST and ImegeNette. The results are a bit mixed. gamma correction seems to work, but for Gaussian blur their method seems to perform worse.

**Strengths:**

1. Enforcing continuous symmetries using infinitesimal generators is in principle a good idea and can make them tractable.
2. Using their method for symmetry discovery, they recover the ground truth generator for gaussian blur.
3. some experimental results on gamma correction in MNISt seem good.

**Weaknesses:**

1. Most of the theory is almost verbatim repetition of Shaw 2024, 2025.
2. The contribution seems to be adapting an existing method to images, but in a very limited way. The diagonal action seems quite restrictive to me and only captures a very small class of symmetries in images. Importantly, it shouldn't be able to handle spatial or steerable symmetries (right?).
3. Experiments are limited. The baseline should have included at least LieGG (Moskalev Neurips 2022), which also uses infinitesimal symmetry regularization, and LieGAN for the symmetry discovery part.
4. Some experimental results, like gaussian blur on MNIST, Table 2, seem to show regularization actually hurts at high noise levels, and dramatically worse than baseline or augmentation. And this is not even symmetry discovery, rather a know symmetry where augmentation is actually possible. If your argument is augmentation would be expensive or must be done many times, this table isn't showing that.

**Questions:**

1. What are the distinguishing theoretical contributions of this work compared to Shaw 2024?
2. Table 2: the fact that your method leads to worse results at high noise, is you method somehow not mixing neighboring pixel data enough? Are you kernels too small? Are they 7x7? That seems quite big.

---

> ### Author Response · Authors · 2025-11-21
> **Initial response**
>
> *Weaknesses*
>
> 1. The background information about flows and vector fields is indeed largely owing to (Shaw et al., 2024 \& 2025). To resolve the reviewer's concern, we will delegate this material to the appendices. However, we feel the inclusion of this material is still of importance for two reasons: (1) it is the theoretical foundation upon which our contribution is built; (2) the concepts of multiparameter group actions, together with their infinitesimal generators, appears largely underexplored in the context of machine learning, owing in part to what seems to be a relatively low citation count for (Shaw et al., 2024 \& 2025): that is to say, these sources do not appear to be well known to the community.
>
> 2. The contribution is nuanced, but it is not limited. It is true that we are discovering and enforcing symmetry in much the same way, at a high-level view, as the methods presented in (Shaw et al., 2024 \& 2025). However, the methods as given in these papers are not well-suited for image data, and our adaptation for image data is not trivial. The simple fact of our reducing/restricting the types of symmetries using diagonal group actions and CNNs is key to why our method can work with images while others can not.
>
> Please see the over-arching comment for a more detailed description of the issues with comparing to LieGAN.
>
> To the question of spatial or steerable symmetries, this is not so "cut-and-dry." The diagonal action concept could apply in two ways: the same group action applied across each color channel, or the same RGB action applied to each pixel. The former case describes Gaussian blur for RGB images, which is "spatial," though qualitatively different than planar transformations such as rotations and translations (which planar transformations we do not consider in this work). For other transformations we consider (such as the power law transformation), the latter description applies: the power law transform applies to each pixel so as not to be spatial. However, symmetry detection can still be accomplished by considering linear combinations of pre-determined infinitesimal generators. The difference is that symmetry detection takes place in 3-dimensions (RGB) instead of $(3 \cdot 32 \times 32)$-dimensions (for CIFAR-10 images). Therefore, like symmetries representable using convolutions, restricting to symmetries representable by diagonal group actions allows us to severely restrict the symmetry search space, but in a way that is suited for common image transformations.
>
>
> 3. The methods of LieGG and LieGAN are entirely unsuited for our experiments, owing to our novel but nuanced contribution. First, these methods are largely limited to affine transformations, whereas image transformations are not generally affine: meanwhile, a transformation as simple as the power law transformation is not affine. (We acknowledge that LieGAN has been extended to handle non-affine symmetries, but the need to employ a larger-dimensional latent space merely compounds the issue described above.) The second reason is the same reason that the methods given in (Shaw et al., 2024 \& 2025) are unsuitable: they do not respect the structure of the images, which also leads to an impractically large symmetry search space, as described above.
>
> 4. We are not trying to outperform augmentation. We are demonstrating a possible implementation of our proposed framework in order to show that symmetry regularization can lead to better model generalization and/or performance, similar to augmentation. But our framework does not require explicit knowledge of the group action, unlike augmentation and many other methods: it needs only infinitesimal generators, thus making use of the efficient symmetry discovery method of using vector fields. Additionally, very different results can be obtained by changing the regularization parameter.
>
>
> *Questions*
>
> 1. The distinguishing theoretical contribution is our tailoring/adapting symmetries to be suitable for image data. Our approach differs from other previous approaches which consider image data in that we consider transformations on the pixel values themselves rather than planar transformations like rotations and shifts: beyond the power law and Gaussian blur transformations, other common transformations include adjusting brightness, contrast, and hue.
>
> 2. There may be a misunderstanding concerning this experiment: we are not considering varying levels of noise. We are assessing the abilities of each model to generalize to progressively blurred images.

---

### Author Response · Authors · 2025-11-21
**Our specialization to image transformations is not a trivial contribution**

Our first general comment relates to the large efficiency gains from our methodology. The suggestion to compare with LieGAN, LieGG, and related methods, coupled with the notion that our contribution offers only a limited improvement to (Shaw et al., 2024 \& 2025) indicates that our paper, as currently written, does not explain the issue at hand explicitly enough. And so, we consider the task of detecting Gaussian blur for MNIST images anew. The Gaussian blur transformation is an affine transformation of the pixel values, since the pixel values are updated to become a linear combination of their neighbors, and so we may consider, in a thought experiment, other approaches. The matrix-based method of LieGAN would require matrices of an astounding size of $784\times 784$ simply to parameterize the space of affine symmetries, on top of the computational overhead of matrix exponentiation, explicitly transforming the images, measuring divergences, and otherwise training a GAN. To put this number of parameters (over $600,000$) into perspective, that is an order of magnitude above the number of training points in the dataset, and closer to two orders of magnitude above the number of parameters in a typical classifier for this data. And the problem would be much worse if we considered ImageNet: the $224 \times 224$ images yield matrix transformations having over 2.5 **billion** entries (not even considering separate RGB channels), which is $1-3$ orders of magnitude larger than the number of parameters needed to train an image classifier (and larger still than the number of training images). Even the method introduced in (Shaw et al., 2024 \& 2025) cannot escape the large search space, despite cutting out much of the computational overhead employed by matrix-based approaches. Meanwhile, the experiment in our paper discovers this symmetry using a mere 49 parameters--learnable weights of a 7x7 kernel. The flexibility in how the components of the vector field infinitesimal generators can be specified allows us to restrict the initially-large search space of symmetries so that only image-specific transformations--such as those accomplished by means of convolutions--can be considered.

---

### Author Response · Authors · 2025-11-21
**An experiment with multiple symmetries**

In response to reviewers a4Zt and sigs, though for the benefit of each reviewer, we consider symmetry enforcement on MNIST using both Gaussian blur and the power law transform. Regularization, Augmentation, and the baseline are each trained for 10 epochs. For augmentation, Gaussian blur is applied before the power law transform, with $\gamma \in \{ 0.25, 1.0 \}$ and $\sigma \in \{ 0.1, 3.0 \}$. For regularization, $\lambda = 0.2$ for the final 8 epochs (it begins as 0). Only one repetition is conducted for each method. Each method is evaluated twice (and has two tables): once for evaluating the power law transform before the Gaussian blur, another for switching that order.

These results show regularization dominating in the "upper right" section--that is, closer to the actual test data--while augmentation maintains more consistent scores throughout. Both methods uniformly outperform the baseline. While regularization and the baseline have lower scores for evaluating Gaussian blur first, the augmentation method flips this trend, probably owing to the fact that the data was augmented by first applying the Gaussian blur transformation before the power law transformation.

**Regularization**

| $\gamma$, $\sigma$   | $\gamma = 0.1$ | $\gamma = 0.2$ | $\gamma = 0.25$ | $\gamma = 0.5$ | $\gamma = 1.0$ |
|-----------|---------------------------|---------------------|---------------------|---------------------|---------------------|
| $\sigma = 0.1$ | $0.9012$ | $0.9662$ | $0.9693$ | $0.9716$ | $0.9718$ |
| $\sigma = 1.0$ | $0.8527$ | $0.9579$ | $0.9642$ | $0.9683$ | $0.9684$ |
| $\sigma = 2.0$ | $0.7869$ | $0.8907$ | $0.9227$ | $0.9445$ | $0.9469$ |
| $\sigma = 3.0$ | $0.7323$ | $0.8170$ | $0.8616$ | $0.9107$ | $0.9158$ |
| $\sigma = 6.0$ | $0.6797$ | $0.7743$ | $0.8016$ | $0.8699$ | $0.8777$ |

| $\sigma$, $\gamma$   | $\gamma = 0.1$ | $\gamma = 0.2$ | $\gamma = 0.25$ | $\gamma = 0.5$ | $\gamma = 1.0$ |
|-----------|---------------------------|---------------------|---------------------|---------------------|---------------------|
| $\sigma = 0.1$ | $0.9012$ | $0.9662$ | $0.9693$ | $0.9716$ | $0.9718$ |
| $\sigma = 1.0$ | $0.7970$ | $0.9183$ | $0.9407$ | $0.9644$ | $0.9684$ |
| $\sigma = 2.0$ | $0.6070$ | $0.7374$ | $0.7854$ | $0.9200$ | $0.9469$ |
| $\sigma = 3.0$ | $0.5326$ | $0.6590$ | $0.7022$ | $0.8626$ | $0.9158$ |
| $\sigma = 6.0$ | $0.4869$ | $0.6027$ | $0.6411$ | $0.7989$ | $0.8777$ |

**Augmentation**

| $\gamma$, $\sigma$   | $\gamma = 0.1$ | $\gamma = 0.2$ | $\gamma = 0.25$ | $\gamma = 0.5$ | $\gamma = 1.0$ |
|-----------|---------------------------|---------------------|---------------------|---------------------|---------------------|
| $\sigma = 0.1$ | $0.9407$ | $0.9463$ | $0.9481$ | $0.9510$ | $0.9463$ |
| $\sigma = 1.0$ | $0.9299$ | $0.9417$ | $0.9455$ | $0.9490$ | $0.9459$ |
| $\sigma = 2.0$ | $0.8989$ | $0.9249$ | $0.9302$ | $0.9397$ | $0.9381$ |
| $\sigma = 3.0$ | $0.8768$ | $0.9110$ | $0.9202$ | $0.9306$ | $0.9298$ |
| $\sigma = 6.0$ | $0.8516$ | $0.8967$ | $0.9072$ | $0.9220$ | $0.9207$ |

| $\sigma$, $\gamma$   | $\gamma = 0.1$ | $\gamma = 0.2$ | $\gamma = 0.25$ | $\gamma = 0.5$ | $\gamma = 1.0$ |
|-----------|---------------------------|---------------------|---------------------|---------------------|---------------------|
| $\sigma = 0.1$ | $0.9407$ | $0.9463$ | $0.9481$ | $0.9510$ | $0.9463$ |
| $\sigma = 1.0$ | $0.9446$ | $0.9501$ | $0.9516$ | $0.9533$ | $0.9459$ |
| $\sigma = 2.0$ | $0.9389$ | $0.9492$ | $0.9506$ | $0.9509$ | $0.9381$ |
| $\sigma = 3.0$ | $0.9288$ | $0.9435$ | $0.9468$ | $0.9479$ | $0.9298$ |
| $\sigma = 6.0$ | $0.9186$ | $0.9395$ | $0.9426$ | $0.9438$ | $0.9207$ |

**Baseline**

| $\gamma$, $\sigma$   | $\gamma = 0.1$ | $\gamma = 0.2$ | $\gamma = 0.25$ | $\gamma = 0.5$ | $\gamma = 1.0$ |
|-----------|---------------------------|---------------------|---------------------|---------------------|---------------------|
| $\sigma = 0.1$ | $0.7958$ | $0.8781$ | $0.8836$ | $0.8883$ | $0.8886$ |
| $\sigma = 1.0$ | $0.6796$ | $0.8641$ | $0.8731$ | $0.8836$ | $0.8855$ |
| $\sigma = 2.0$ | $0.4252$ | $0.7673$ | $0.8018$ | $0.8449$ | $0.8533$ |
| $\sigma = 3.0$ | $0.3864$ | $0.6929$ | $0.7382$ | $0.7980$ | $0.8106$ |
| $\sigma = 6.0$ | $0.3737$ | $0.6168$ | $0.6841$ | $0.7482$ | $0.7636$ |

| $\sigma$, $\gamma$   | $\gamma = 0.1$ | $\gamma = 0.2$ | $\gamma = 0.25$ | $\gamma = 0.5$ | $\gamma = 1.0$ |
|-----------|---------------------------|---------------------|---------------------|---------------------|---------------------|
| $\sigma = 0.1$ | $0.7958$ | $0.8781$ | $0.8836$ | $0.8883$ | $0.8886$ |
| $\sigma = 1.0$ | $0.5542$ | $0.7908$ | $0.8254$ | $0.8726$ | $0.8855$ |
| $\sigma = 2.0$ | $0.3600$ | $0.5933$ | $0.6659$ | $0.7889$ | $0.8533$ |
| $\sigma = 3.0$ | $0.3368$ | $0.5107$ | $0.5778$ | $0.7382$ | $0.8106$ |
| $\sigma = 6.0$ | $0.3270$ | $0.4643$ | $0.5220$ | $0.6945$ | $0.7636$ |

---

### Meta-Review · Area_Chair_iREB · 2025-12-28

**Summary:**

While recognizing that the paper makes a number of interesting contributions, there were a significant number of reviewer concerns raised:
- This builds extensively on existing work (Shaw 2024, 2025 have been mentioned by multiple reviewers), and the contributions beyond these are limited, or have not clearly sold (V48H W1 Q1, sigs W3, SbJc W para1).
- Limited novelty and/or narrow contributions (V48H W2, a4Zt W5,6, sigs W3, SbJc W paras1-2).
- Limited experiments especially in comparing to existing baselines (V48H W3, a4Zt W2,4 Q5, sigs W2, SbJc W para 3).
- Experimental results were poor / unconvincing, (V48H W4 Q2, a4Zt W3 Q3, sigs W1 Q1).
- Incomplete derivation / investigation (a4Zt W1 various Q's).

In subsequent instances with reviewer responses:
- Reviewer sigs raised the point that augmentation still appeared a simpler and more general solution, and requested a numerical evaluation of the usefulness of the proposed approach under conditions in which augmentations are not computationally feasible. The AC interprets this as requesting the authors to point out concrete examples of such scenarios, and undertake computational cost analysis.
- Reviewer SbJc reiterated that the methodological framework was not clearly described, which would require three important components to be developed further (as detailed in their comment).

**Reviewer Concerns:**

The authors undertook to provide responses that were quite substantial in nature, including additional responses. However, the very extensive scope of concerns across all four reviewers meant that the responses could only partially addressed these concerns. As further evidence, in the only two reviewer responses, both sigs and SbJc indicated that they were not going to raise their scores.

The AC, in keeping with some of the reviewer comments, acknowledges that there is some potentially interesting contributions that the authors have put forth in their paper. However, there remains much work to be done, both in terms of improving the exposition of the paper, and additional experiments, before this work is ready for reconsideration for acceptance.

**Reviewer Scores:**

The reviewer scores are currently two 2's, and two 4's. The AC does not expect the reviewer scores to improve under a normal reviewer discussion process, given the substantial nature of concerns raised and the scale of improvements that have been requested by the reviewers.

---

### Decision · Program_Chairs · 2026-01-26

Reject